# Anti-Fatigue Effect of a Dietary Supplement from the Fermented By-Products of Taiwan Tilapia Aquatic Waste and *Monostroma nitidum* Oligosaccharide Complex

**DOI:** 10.3390/nu13051688

**Published:** 2021-05-15

**Authors:** Ying-Ju Chen, Chun-Yen Kuo, Zwe-Ling Kong, Chin-Ying Lai, Guan-Wen Chen, An-Jen Yang, Liang-Hung Lin, Ming-Fu Wang

**Affiliations:** 1Bachelor Program in Health Care and Social Work for Indigenous Students, College of Humanities & Social Sciences, Providence University, Shalu Dist., Taichung 43301, Taiwan; yjchen5@pu.edu.tw (Y.-J.C.); ajyang@pu.edu.tw (A.-J.Y.); 2Department of Social Work and Child Welfare, Providence University, Shalu Dist., Taichung 43301, Taiwan; cykuo2@pu.edu.tw; 3Department of Food Science, National Taiwan Ocean University, No.2, Pei-Ning Road, Jhongjheng Dist., Keelung 20224, Taiwan; kongzl@mail.ntou.edu.tw (Z.-L.K.); chengw@mail.ntou.edu.tw (G.-W.C.); 4Master Program in Social Enterprises & Cultural Creativity, College of Humanities & Social Sciences, Providence University, Shalu Dist., Taichung 43301, Taiwan; cylai1@pu.edu.tw; 5Department of Food and Nutrition, Providence University, Shalu Dist., Taichung 43301, Taiwan; 6Division of Allergy, Immunology & Rheumatology, Taichung Tzu Chi Hospital, Buddhist Tzu Chi Medical Foundation, Tanzi Dist., Taichung 42743, Taiwan

**Keywords:** Tilapia, by-product, *Monostroma nitidum*, oligosaccharide, anti-fatigue, exercise performance, SAMP8

## Abstract

The Taiwan Tilapia is an important aquaculture product in Taiwan. The aquatic by-products generated during Tilapia processing, such as fish bones and skin, are rich in minerals and protein. We aimed to explore the effect of a dietary supplement, comprising a mixture of fermented Tilapia by-products and *Monostroma nitidum* oligosaccharides as the raw materials, combined with physical training on exercise performance and fatigue. We used a mouse model that displays a phenotype of accelerated aging. Male senescence-accelerated mouse prone-8 (SAMP8) mice were divided into two control groups—with or without physical training—and supplemented with different doses (0.5 times: 412 mg/kg body weight (BW)/day; 1 time: 824 mg/kg BW/day; 2 times: 1648 mg/kg BW/day) of fermented Tilapia by-products and *Monostroma nitidum* oligosaccharide-containing mixture and combined with exercise training groups. Exercise performance was determined by testing forelimb grip strength and with a weight-bearing exhaustive swimming test. Animals were sacrificed to collect physical fatigue-related biomarkers. Mice dosed at 824 or 1648 mg/kg BW/day showed improvement in their exercise performance (*p* < 0.05). In terms of biochemical fatigue indicators, supplementation of 824 or 1648 mg/kg BW/day doses of test substances could effectively reduce blood urea nitrogen concentration and lactate concentration and increase the lactate ratio (*p* < 0.05) and liver glycogen content post-exercise (*p* < 0.05). Based on the above results, the combination of physical training and consumption of a dietary supplementation mixture of fermented Tilapia by-products and *Monostroma nitidum* oligosaccharides could improve the exercise performance of mice and help achieve an anti-fatigue effect.

## 1. Introduction

Studies have shown that fatigue is defined as a negative impact on work, athletic performance, family life, and social relationships due to mental workload and physical fatigue [1]. Energy depletion, excess metabolite accumulation, and oxidative stress are considered important factors in causing physical fatigue [2]. Physical fatigue is usually defined as a reversible reduction in performance during exercise [3,4]. The known fatigue mechanism is related to the availability of metabolic fuel and waste accumulation. Prolonged intense physical exercise will generate reactive oxygen species (ROS), which may cause tissue damage and oxidative stress [5]. During exercise, the body consumes a lot of energy-carrying molecules, such as ATP (Adenosine triphosphate), phosphocreatine, muscle glycogen, etc. When the intensity and period of exercise increase, the body’s oxygen consumption also increases. Sports fatigue occurs when the body does not get timely energy supplements. After prolonged or intense exercise, the tissue cells accelerate anaerobic glycolysis due to severe hypoxia. The amount of lactate produced in the muscle increases and slowly accumulates, causing a rise in hydrogen ion concentration in the body, a subsequent decrease in blood pH, and the acceleration of fatigue [6]. Oxygen uptake and electron transport chain reaction increase during exercise, related to reactive oxygen species (ROS) production. Active oxygen species gradually accumulate and cause harm by excess oxygen radicals damaging human cells, protein, fat, and DNA [7,8]. In addition, Zheng et al. found that exogenous antioxidants can reduce the oxidative stress caused by exercise [9]. To enhance exercise performance, nutritional supplementation with carbohydrates and amino acids, particularly branched-chain amino acids (BCAAs), is a critical factor. In recent years, the study of BCAAs has helped promote cell metabolism, thereby enhancing and restoring exercise performance.

Tilapia is an important aquaculture product in Taiwan. Approximately 60–75% of the waste generated from fish processing [10], such as fish bones, skin, and other by-products, is estimated to weigh up to 4500 metric tons. Fishbones account for about 15–20% of the fish; they are rich in protein, vitamins, minerals, trace elements, DHA (Docosahexaenoic Acid), and other nutrients. To improve the overall economic benefits of high-quality Wu-Guo Taiwanese carp (Tilapia), the remaining fish bones and skin by-products from the processing of Tilapia slices were refined using an advanced extraction and concentration technology and a lactate bacterial fermentation technology. The fermentation product was mixed with *Monostroma nitidum* oligosaccharides to form a new complex material for health care. The mixture of fermented Tilapia by-products and the *Monostroma nitidum* oligosaccharides is rich in branched-chain amino acids (BCAAs), including leucine, isoleucine, and valine. The average total BCAA content of the functional component of the compound of fermented by-products of Taiwan Tilapia aquatic waste and *Monostroma nitidum* oligosaccharide complex is 218 mg/g. 

Moderate exercise or physical training is beneficial for enhancing cardiopulmonary function, improving exercise performance, and helping to prevent the development of chronic diseases [11]. This study used an accelerated aging mouse model to investigate the impact of physical training combined with dietary supplementation using the BCAA-rich mixture of fermented Tilapia by-products and *Monostroma nitidum* oligosaccharides on exercise performance and fatigue.

## 2. Materials and Methods

### 2.1. Material Preparation

#### 2.1.1. Material Origin

By-products of Taiwan Tilapia (*Oreochromis niloticus*), which include the remaining fishbone, residual meat and tail after removing the body, head, and internal organs, were procured from Fortune Life Enterprise Co., Ltd. (Kaohsiung City, Taiwan). *Monostroma nitidum* was bought from Penghu Seafood House (Penghu, Taiwan) and was stored at −20 °C for preparation.

#### 2.1.2. Hot Extraction Process of Fish Bone Powder

The fishbone frame was whipped into a muddy form, and distilled water (solid:liquid ratio = 1:5) was added. The mixture was boiled to 4 °Brix (Degrees Brix, a unit of sucrose measurement in a liquid). The clear liquid and fishbone residue were separated and extracted. The clear liquid was boiled to 8 °Brix and labeled as liquid A. The fishbone residue with 5 L distilled water was boiled to 3–3.5 °Brix and then filtered and labeled as liquid B. Liquid A and liquid B (°Brix came to 6–6.5) were mixed and stored at −20 °C for freezing and degreasing. After freezing and drying the mixture (in FreeZone^®^ 12 L Console Freeze Dryers, Labconco^®^, Kansas City, MO, USA), fishbone powder was obtained using this procedure.

#### 2.1.3. Preparation of Fish Bone Powder Fermentation Broth

The method of fermenting fish bone powder with lactate bacteria is referenced and modified from Chen et al. [12]. The fishbone broth was supplemented with 1% lactic bacteria (*Enterococcus faecalis*, BCRC14046 and *Lactobacillus rhamnosus*, BCRC10940 after secondary activation and cultured to OD_600_ = 0.8–1.0) and fermented at 37 °C for 48 h. After fermentation was completed, the fishbone powder broth was placed in an autoclave (Tomy Seiko, Tokyo, Japan) and heated at 121 °C for 15 min to terminate the fermentation. After the fermentation liquid was cooled to room temperature, it was centrifuged at 14,000× *g* for 30 min. The supernatant was filtered with a 0.45 μm membrane filter (Finetech Research and Innovation Co., Taichung, Taiwan) under an air suction filtration device.

#### 2.1.4. Preparation of *Monostroma Nitidum* Polysaccharide Hydrolysates

The polysaccharide hydrolysate was prepared from *Monostroma nitidum* by enzyme hydrolysis using the protocol provided by Wu et al. [13,14]. Hot extraction fluid of *Monostroma nitidum* was extracted from *Monostroma nitidum* by adding distilled water up to 10% (w/v) and heating at 121 °C for 30 min. MMB (Modified Marine Broth with Algal powder) medium was made by adding 1.5% (*w*/*v*) *Monostroma nitidum* and 1.5% (*w*/*v*) hot extraction fluid of *Monostroma nitidum.* Under an aseptic environment (vertical floor type cabinet BSC-3, Chih Chin H&W Enterprise Co., Ltd., Taipei, Taiwan), we inoculated 1% (*w*/*v*) activated MA103 (*Pseudomonas vesicularis* MA103) and MAEF108 (*Aeromonas salmonicida* MAEF108) to 13 mL MMB medium; it was shaken for 48 h at 26 °C to induce enzyme formation from MA103 and MAEF108. The enzyme solution was centrifuged at 14,000× *g* for 30 min (CR-21G, Hitachi Co., Ltd., Tokyo, Japan). The endotoxin was removed by a 10 kDa filter. Crude enzyme extract was obtained for the clarified liquid by filtering with a 0.22 μm filter membrane. Then, 10% (*v*/*v*) MA103 and MAEF108 crude enzyme extracts were added to the hot extraction fluid of *Monostroma nitidum*, and the mixture was shaken and hydrolyzed at 37 °C for 48 h. The enzyme reaction was then stopped by heating at 121 °C for 20 min, and finally it was freeze-dried into powder. The fermented fish bone powder and *Monostroma nitidum* polysaccharide hydrolysate powder were mixed at a 1:2 ratio for the sample of subsequent analysis and to feed test animals.

### 2.2. Test Animals and Study Design

Senescence-accelerated mouse prone-8 (SAMP8) was used as the animal model for this study. The study protocol was carried out in strict accordance with the recommendations in the Guide for the Care and Use of Laboratory Animals of the National Institutes of Health and approved by the Animal Research Ethics Committee at Providence University (IACUC number 20161205-A02). The animals were housed in a transparent plastic cage of 30 (W) × 20 (D) × 10 (H) cm^3^. The temperature and relative humidity of the animal room were maintained at 25 ± 2 °C and 65 ± 5%, respectively, and it was a dust-free automatic control room. The lighting cycle was controlled by an automatic timer. A total of 75 four-month-old male SAMP8 mice were selected as experimental animals and randomly divided into 5 groups of 15 mice. Group A (Gr-A) was a blank control without exercise. Group B (Gr-B) was a blank control with exercise. Groups C, D, and E received three different doses of supplementation with a mixture of fermented Tilapia by-products and *Monostroma nitidum* oligosaccharide complex combined with exercise training: 0.5 dose—412 mg/kg BW/day (Gr-C), 1 dose—824 mg/kg BW/day (Gr-D), 2 dose—1648 mg/kg BW/day dose (Gr-E). The experimental mice were free to take food and water. The body weight of mice, food intake amount, and water intake volume were recorded. The control group was fed ddH_2_O (double-distilled H_2_O); all other experimental groups of mice were tube-fed samples. Experimental designed exercise-mice groups (Gr-A, Gr-B, Gr-C, Gr-D, Gr-E) were trained, and forced-swimming exercise mode was carried out weekly for 6 weeks. Subsequently, the following tests were performed: (1) forelimb grip strength [15], (2) weight-bearing (5% body weight) exhaustive swimming test, (3) blood biochemical values and fatigue-related biomarkers.

### 2.3. Swimming Exercise Training Mode

The training model for swimming exercise was carried out with reference to the study by Elia et al. [16]. The tap water pool temperature was set at 27 ± 1 °C, and the water level was set at 40 cm in depth. Progressive enhanced endurance swimming training mode was followed according to the ET (exercise training) protocol described in a study by Chen et al. [17].

### 2.4. Forelimb Grip Strength Test

The test was carried out 30 min after the last sample feed; an animal forelimb grip strength measuring device (Ugo Basile Grip Strength Meter, Cat. No. 47200, Gemonio, Italy) was used. The test animal was placed on the experimental platform, and the distal one-third of the mouse tail was grabbed. The mouse was allowed to grasp the measuring device, and then the mouse was pulled in the opposite direction in parallel. This was repeated 3 times, and the maximum value was recorded. When the animal’s tail is dragged, the animal instinctively grabs the front bar to prevent unintended backward movement until the operator pulls more than its maximum grip. When the animal releases the grab-bar, the instrument automatically records the maximum force value [15].

### 2.5. Exhaustive Swimming Test

The test [17] was carried out 30–60 min after the last sample feed. The mice were placed in a water tank of 15 cm in diameter and 20 cm in water depth, forcing them to swim until they were physically exhausted. At least 12 h of fasting was required before the exhaustive swimming test. The test was carried out one mouse at a time. In order to shorten the test time, a lead wire (fuse) was attached to the back of the mouse for weight-bearing swimming. The bearing-weight was 5% of the animal’s body weight. During the test, the water temperature was controlled at 27 ± 1 °C, and an appropriate amount of surfactant was added. The limbs of the test animals were kept moving throughout the test. If the test animal floated on the water surface and did not move, a stir bar was used to stir the water nearby. The total time from when the mouse was put into the water until it kept its head below water level for at least 8 s was recorded as the exhaustive swimming time.

### 2.6. Biochemistry Tests

Assay kits used for the determination of hepatic glycogen level (Abcam ab65620) and lactate (Abcam colorimetric ab65331) were purchased from Biochiefdom International Co., Ltd. (New Taipei City, Taiwan). Glucose, total protein, albumin, triglycerides, total cholesterol, alkaline phosphatase (ALP), glutamate oxaloacetate transaminase (GOT), glutamate pyruvate transaminase (GPT), blood urea nitrogen (BUN), and creatine kinase (CK) levels were tested. All other reagents used in this study were of analytical grade.

### 2.7. Analysis of Fatigue-Related Biomarkers

#### 2.7.1. Blood Urea Nitrogen Test

After the last feed (duration: 30 min), mice swam in the water at a temperature of 30 °C for 90 min without weight-loading, and blood samples were collected after a 60 min rest period. For the sake of humanity, the experimental animals were not treated excessively in the experiment. We followed a test procedure used in a previously published article [18]. Using the colorimetry method, urease was added to the quantitative plasma or serum. After the previous reaction, ammonia was added as a coloring agent. The red compound was produced after the reaction, and the absorbance was measured at a wavelength of 660 nm. The BUN concentration was then calculated.

#### 2.7.2. Hepatic Glycogen Test

After the last feed of 30 min, the animals were sacrificed, and the liver was isolated and homogenized in 10% solution with normal saline at 4 °C. The glycogen was decomposed into glucose, and the concentration was measured using available kits.

#### 2.7.3. Blood Lactate Test

Blood was collected 30 min after the last feed. Mice swam without weight-loading in 30 °C temperature water for 10 min. Then, blood was collected again after a 20 min rest period post-swimming. The drawn blood underwent an enzymatic reaction and colorimetry analysis. Lactate oxidase was added to the quantitative plasma, then 4-aminoantipyrine and 1,7-dihydroxynaphthalene were added, and a red compound was produced by peroxidase. Absorbance was measured at a wavelength of 540 nm, and the concentration of lactate at that absorbance was calculated.

### 2.8. Statistical Analysis

The data obtained in this study were statistically analyzed using SPSS statistical software package (v19.0, IBM., Armonk, NY, USA). The values of the experimental results were presented as mean ± standard error (mean ± SEM). The experimental data were analyzed by one-way analysis of variance (one-way ANOVA) to determine the differences between groups, and Duncan’s multiple range test was used to compare differences between groups. A *p* < 0.05 was considered to indicate a significant difference.

## 3. Results

### 3.1. Impact of Body Weight, Intake Amount, and Water Consumption Volume After Six Weeks of Supplementation with a Complex of Fermented Tilapia By-Products and Monostroma Nitidum Oligosaccharides Combined with Physical Training

Table 1 demonstrates the differences between body weight, daily average food intake, and daily water consumption of male SAMP8 mice fed with a mixture of Tilapia by-product fermentation and *Monostroma nitidum* oligosaccharides combined with exercise training for six weeks. To minimize experimental bias, mice were randomly divided into groups; there were no significant differences in the initial body weight in each group.

The results showed no significant difference in body weight, food intake, and water consumption in the blank-no-training group, blank-training group, and experimental groups before and after tests. No impact was observed on the body weight, daily average food intake, and daily water consumption in male SAMP8 mice fed with supplementation of a mixture of fermented Tilapia by-products and *Monostroma nitidum* oligosaccharides combined with exercise training.

### 3.2. Evaluation of Exercise Performance After Six Weeks of Supplementing with a Mixture of Fermented Tilapia By-Products and Monostroma Nitidum Oligosaccharides Combined with Exercise Training

Time of swimming to exhaustion is a commonly used indicator of reaction exercise ability. ET can significantly increase absolute and relative grip strength. In animals of different ages and different species, physical training for specific exercises enhanced both forelimb grip strength and exhaustive swimming time (EST) [17,19]. Another study pointed out that the forelimb grip strength positively correlated with muscle strength [20]. In this study, the forelimb grip strength and EST were used to evaluate the impact of exercise performance and fatigue on mice supplemented with a mixture of fermented Tilapia by-products and *Monostroma nitidum* oligosaccharides combined with ET.

Figure 1 demonstrates the forelimb grip strength results for male SAMP8 mice supplemented with the mixture of fermented Tilapia by-products and *Monostroma nitidum* oligosaccharides combined with ET. Forelimb grip strength was found to be greater in the Gr-D and Gr-E groups than in the Gr-A group (*p* < 0.05). Figure 2 shows the exhaustive swimming test results for male SAMP8 mice that were supplemented with the mixture of fermented Tilapia by-products and *Monostroma nitidum* oligosaccharides combined with ET. All groups combined with training (Gr-B, Gr-C, Gr-D, Gr-E) showed increased ESTs compared with the group Gr-A, with significantly longer ESTs in the Gr-D and Gr-E groups (*p* < 0.05).

### 3.3. Biochemical Blood Analysis

Table 2 exhibits the value changes of blood glucose, total protein, albumin, triglycerides, total cholesterol (CHOL), alkaline phosphate (ALP), glutamate oxaloacetate transaminase (GOT), and glutamate pyruvate transaminase (GPT) after SAMP8 mice were fed with fermented Tilapia by-product and *Monostroma nitidum* oligosaccharide complex combined with ET for six weeks. The results showed no significant difference among these groups, indicating that feeding with fermented Tilapia by-product and *Monostroma nitidum* oligosaccharide complex combined with ET did not adversely affect blood lipid levels or cause dysfunction of the liver and kidney.

### 3.4. Fatigue-Related Biomarker Analysis

#### 3.4.1. BUN Concentration

Figure 3 illustrates the blood urea nitrogen (BUN) concentration of SMAP8 mice fed with Tilapia by-product fermentation and *Monostroma nitidum* oligosaccharide complex combined with ET for six weeks. The results showed that the BUN concentration of the training group was significantly lower than that of the non-training group (Gr-A) (*p* < 0.05) after supplementing with the mixture of fermented Tilapia by-products and *Monostroma nitidum* oligosaccharide complex.

#### 3.4.2. Glycogen Level

Figure 4 displays the liver glycogen level of SMAP8 mice fed with Tilapia by-product fermentation and *Monostroma nitidum* oligosaccharide complex combined with ET for six weeks. The results showed that one (Gr-D) and two (Gr-E) doses of Tilapia by-product fermentation and *Monostroma nitidum* oligosaccharide complex combined with ET could significantly increase the liver glycogen level compared with the non-training group (Gr-A) (*p* < 0.05).

#### 3.4.3. Blood Lactate Level

Table 3 indicates the effect of the exercise test on lactate in SAMP8 mice fed with the mixture of fermented Tilapia by-products and *Monostroma nitidum* oligosaccharide complex combined with exercise training (ET). We refer to the statistical methods of the following documents, and use a statistical method of one-way ANOVA to observe the changes in the lactate level between each group. There was no significant difference observed in blood lactate level between these groups before the exercise test. After the exercise test, the lactate concentration was significantly lower in group Gr-B, group Gr-C, group Gr-D, and group Gr-E than group Gr-A (*p* < 0.05). After the exercise test, the lactate concentration was significantly lower in the Gr-B, Gr-C, Gr-D, and Gr-E groups than in the Gr-A group, by 20.4%, 25.2%, 25.7%, and 27.9% (all *p* < 0.05), respectively. After resting for 20 min after the exercise test, there was no significant difference in blood lactate concentration between the four ET groups (Gr-B, Gr-C, Gr-D, Gr-E); levels remained lower than group for Gr-A. For the serum lactate production rate, the training groups (Gr-B, Gr-C, Gr-D, Gr-E) were significantly lower than the group Gr-A clearance rate of serum lactate; there was no significant difference between all groups. The results showed that in all the study groups that received exercise training (Gr-B, Gr-C, Gr-D, Gr-E), the serum lactate level was lower and the clearance rate of serum lactate was higher. Groups without exercise training did not show such results. This indicates that giving the mixture of Tilapia by-product fermented with *Monostroma nitidum* oligosaccharide combined with physical exercise can slow down the formation of lactate. 

## 4. Discussion

The authors fermented by-products of Taiwan Tilapia aquatic waste and marine green alga *Monostroma nitidum* oligosaccharides complex containing branched-chain amino acids (BCAAs), leucine, isoleucine, and valine. Research has shown that BCAA supplementation prior to squat exercise reduces catabolism in skeletal muscle, decreases delayed-onset muscle soreness (DOMS), and contributes to muscle protein synthesis [21]. When mice were fed with BCAA-containing chicken essence for four weeks, their forelimb grip strength and exhaustive swimming test time were significantly improved [22]. After supplementation treatment with BCAA-rich sake protein and power exercise training (PET) for four weeks, mice had significantly higher grip strength and exhaustive swimming time [23]. These studies showed that supplementation with BCAA helped enhance mice exercise performance, consistent with this study’s results. In summary, ET combined with supplementation by fermented Tilapia by-products and marine green alga *Monostroma nitidum* oligosaccharides complex can enhance exercise performance, as observed with the increase in EST (Figure 2) and increase in forelimb grip strength in SAMP8 mice (Figure 1).

Regulation of blood glucose homeostasis plays a vital role in long-term endurance exercise. Hypoglycemia inhibits brain function during exercise and causes an inability to continue the exercise. Therefore, blood glucose is a crucial fatigue-related blood biochemical indicator [24,25]. Another study has shown that if muscle glycogen and blood sugar are utilized well during exercise, the performance improves, and blood lactate production is reduced post-exercise [26]. Studies have shown that physical training helps regulation of blood sugar levels [19]. Moreover, mice fed with BCAA-rich sake lees for four weeks showed a reduction in the decline in exercise performance caused by exercise-induced hypoglycemia [22]. This study showed higher blood sugar levels in mice on supplementing their diet with different doses of the mixture of fermented Tilapia by-products and *Monostroma nitidum* oligosaccharide complex combined with ET than in the non-training group (Gr-A) (Table 2). Therefore, it is speculated that supplementation with the mixture combined with ET can slow the occurrence of exercise-induced hypoglycemia (Table 2).

Studies have shown that high-intensity training can cause physiological and biochemical tissue damage, such as skeletal muscle damage and muscle cell necrosis (Warren et al., 2001). When muscles are damaged, muscle cells release creatine kinase (CK) into the blood; CK can be an important indicator of muscle damage. Kim et al. randomized 26 college students into experimental and placebo groups; BCAAs and a placebo were administered before the flywheel exercise. Results showed that supplementation with BCAAs reduced serum muscle enzyme activity associated with muscle damage and reduced muscle damage resulting from exhaustion from physical exercises [27]. Other studies have shown that ET can increase blood CK level, which can be reduced in mice by feeding them BCAA-rich whey protein [17]. In this study, blood CK level was higher in the Gr-B group, but there was a tendency for CK level to decrease in the supplementation group (Gr-C, Gr-D, Gr-E) (Table 2).

Studies have shown that exercise duration and intensity affect the changes in BUN concentration [28]. BUN is mainly produced by the catabolism of protein and amino acid. In protein decomposition and energy production, pyruvic acid and a large amount of ammonia are produced by deamination. The ammonia must be metabolized into urea by the hepatic urea cycle and excreted from the kidney by the blood circulatory system. Therefore, when the energy metabolism balance is destroyed during a lengthy or highly intense exercise session, the blood ammonia concentration and the BUN concentration increase. Chen et al. found that after administering the mice BCAA-rich whey protein combined with power exercise swimming training, the BUN concentration in the ET control group and the ET plus whey protein supplementation group was significantly lower than that the non-ET group [17]. This result was similar to the results of this experiment; the BUN concentration after swimming exhaustion exercise can be reduced by supplementing with fermented Tilapia by-products and *Monostroma nitidum* oligosaccharide complex combined with ET (Figure 3).

In the human body, glycogen is mainly stored in the liver and muscle. During exercise, the energy demand increases due to muscle contraction; the blood sugar level decreases and stimulates glycogen decomposition to maintain energy supply. The liver glycogen is mainly used to provide sugar and maintain physiological balance during physical exercises. Studies have shown that glycogen depletion has a high correlation with physical fatigue. Reduction in glycogen use or an increase in glycogen storage can improve exercise endurance capacity [29]. After prolonged or high-intensity exercise, the storage of muscle glycogen is markedly reduced. The high glycogen storage level in the muscle is beneficial for glycolytic action during exercise and improving exercise endurance [30]. In one study, rats trained for six weeks using swimming exercise were randomly divided into three groups: one received a placebo, another received BCAAs, and the last group received leucine (LEU) for one week. After the seventh week of the swimming exhaustion test, the liver glycogen amount in the BCAA group and the LEU group was significantly higher than in the placebo group [31]. We obtained similar results for liver glycogen level. In SAMP8 mice fed with the mixture of fermented Tilapia by-products and *Monostroma nitidum* oligosaccharides complex combined with physical training, the liver glycogen level was elevated, with enhancement in exercise performance and anti-fatigue effects (Figure 4).

Certain biochemical parameters have been published in the journal literature in the past to assess the degree of fatigue and post-exercise injury caused by exercise, such as lactate, BUN, CK, ammonia, and free fatty acids [3,32]. Blood lactate is derived from the anaerobic metabolism of glucose during exercise [33]. Lactate is a metabolite formed by the anaerobic glycolysis of glucose, glycogen in muscle and liver. During rest, lactate production in the body is low; under intense or long-time exercise, lactate production increases because of higher oxygen consumption and accelerates anaerobic metabolism. When the lactate production rate is higher than the removal rate, it results in lactate accumulation (OBLA, the onset of blood lactate accumulation) [34]. Lactate accumulation reduces the activity of phosphate fructose kinase, reduces glycolysis, and reduces the pH of muscle cells due to hydrogen ion dissociation, affecting the release of calcium ions from the sarcoplasmic reticulum and reducing muscle contraction leading to fatigue [6]. Studies have shown that proper utilization of muscle glycogen and blood sugar during exercise can reduce lactate production, which can improve exercise performance and endurance capacity [26]. Past studies have shown that mice with different ET types have lower blood lactate concentrations in swimming tests than in non-trained control groups [19,35]. Our study showed that the supplementation mixture combined with training exercise could also reduce lactate production after exercise and achieve an anti-fatigue effect (Table 3). Our results suggest that the combination of physical training and consumption of dietary supplementation mixture of fermented Tilapia by-products and *Monostroma nitidum* oligosaccharides is an effective method for anti-fatigue. We suggest that this could be a new sport nutrition supplement for reducing fatigue. Therefore, further research on the anti-fatigue effect of some micronutrients in Tilapia aquatic waste, such as DHA, will further address its physiological role and mechanism in our next study.

## 5. Conclusions

In conclusion, after undergoing physical training to perform exercises, the SAMP8 male mice showed improvement in exercise performance and physical fatigue-related biomarkers, but these values were not significantly different from those in the blank control group. After combining physical training with the supplementation mixture of Tilapia fermentation by-product and *Monostroma nitidum* oligosaccharide for six weeks, the liver glycogen storage significantly increased, the amount of lactate after exercise decreased, the elevated lactate ratio decreased, and the BUN concentration was reduced. The EST increased, grip strength of the forelimb increased, exercise performance improved, and an anti-fatigue effect was achieved.

## Figures and Tables

**Figure 1 nutrients-13-01688-f001:**
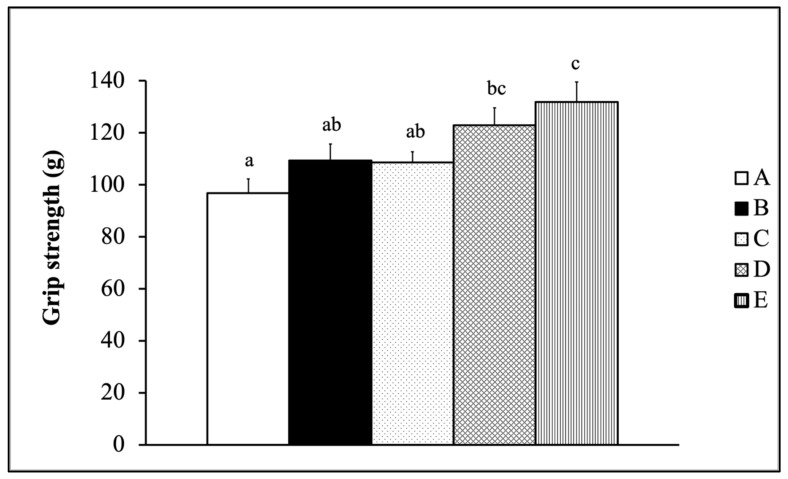
Impact on forelimb grip strength of mice supplemented with the mixture of fermented Tilapia by-products and *Monostroma nitidum* oligosaccharide complex combined with physical training (data are expressed as mean ± S.E.M. (*n* = 15) and analyzed by one-way ANOVA. Bars with different letters (a, b, c) are significantly different at *p* < 0.05. A: Gr-A, B: Gr-B, C: Gr-C, D: Gr-D, E: Gr-E).

**Figure 2 nutrients-13-01688-f002:**
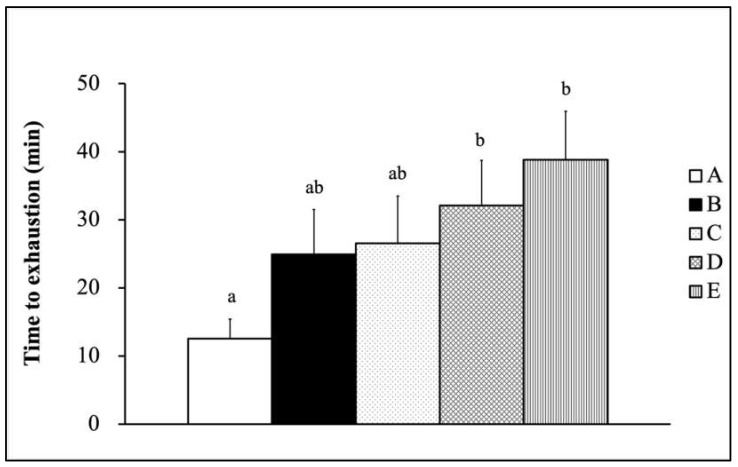
Exhaustive swimming time of mice supplemented with the mixture of fermented Tilapia by-products and *Monostroma nitidum* oligosaccharide complex combined with physical training (data are expressed as mean ± S.E.M. (*n* = 15) and analyzed by one-way ANOVA. Bars with different letters (a, b) are significantly different at *p* < 0.05. A: Gr-A, B: Gr-B, C: Gr-C, D: Gr-D, E: Gr-E).

**Figure 3 nutrients-13-01688-f003:**
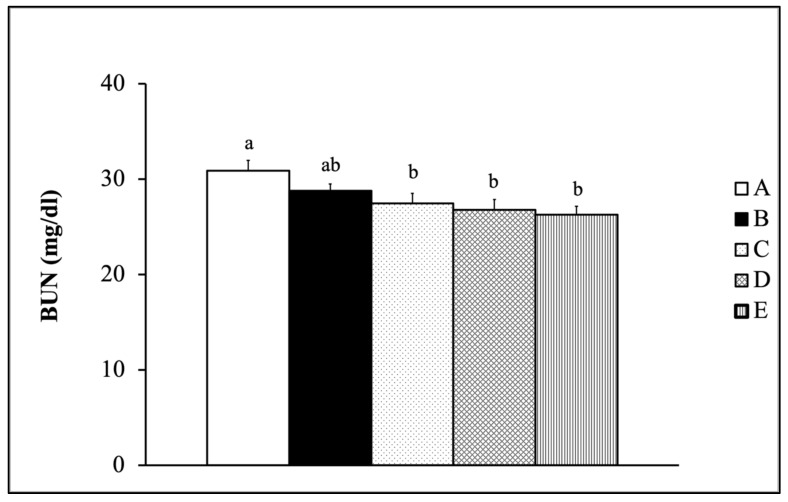
Blood urea nitrogen (BUN) concentration of SMAP8 mic fed with fermented Tilapia by-products and *Monostroma nitidum* oligosaccharide complex combined with physical training (data are expressed as mean ± S.E.M. (*n* = 15) and analyzed by one-way ANOVA. Bars with different letters (a, b) are significantly different at *p* < 0.05.A: Gr-A, B: Gr-B, C: Gr-C, D: Gr-D, E: Gr-E).

**Figure 4 nutrients-13-01688-f004:**
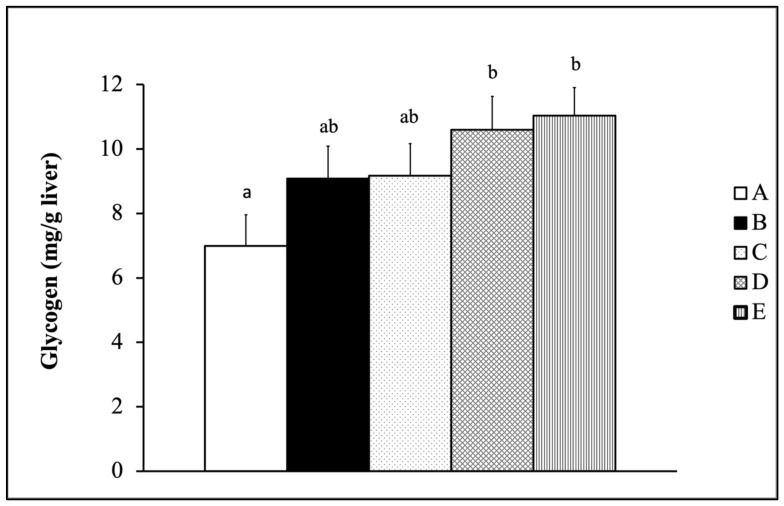
Liver glycogen level of SMAP8 mice fed with Tilapia by-product fermentation and *Monostroma nitidum* oligosaccharide complex combined with exercise training (data are expressed as mean ± S.E.M. (*n* = 15) and analyzed by one-way ANOVA. Bars with different letters (a, b) are significantly different at *p* < 0.05. A: Gr-A, B: Gr-B, C: Gr-C, D: Gr-D, E: Gr-E).

**Table 1 nutrients-13-01688-t001:** Body weight, intake amount, and water consumption volume of SAMP8 mice fed with a mixture of fermented Tilapia by-products and *Monostroma nitidum* oligosaccharides combined with ET.

Group (*n* = 15)	Body Weight (g)		Food Intake (g/Day)	Water Consumption (mL/Day)
	Initial	Final		
A	30.30 ± 0.44	31.55 ± 0.57	5.74 ± 0.14	6.69 ± 0.13
B	29.68 ± 0.68	32.09 ± 0.42	5.54 ± 0.13	6.93 ± 0.10
C	29.94 ± 0.51	31.27 ± 0.52	5.31 ± 0.09	7.15 ± 0.11
D	29.20 ± 0.43	31.21 ± 0.51	5.47 ± 0.18	6.95 ± 0.12
E	30.05 ± 0.46	31.94 ± 0.45	5.23 ± 0.15	7.10 ± 0.13

Notes: Data are expressed as mean ± S.E.M. (*n* = 15) and analyzed by one-way ANOVA. Data were not significantly different (*p* > 0.05) from each other according to ANOVA. A: blank control without exercise (Gr-A). B: blank control with exercise (Gr-B). C: 412 mg/kg BW/day fermented Tilapia by-product and *Monostroma nitidum* oligosaccharide complex combined with exercise training (ET) (0.5 times dose) (Gr-C). D: 824 mg/kg BW/day fermented Tilapia by-product and *Monostroma nitidum* oligosaccharide complex combined with ET (1 times dose) (Gr-D). E: 1648 mg/kg BW/day fermented Tilapia by-product and *Monostroma nitidum* oligosaccharide complex combined with ET (2 times dose) (Gr-E).

**Table 2 nutrients-13-01688-t002:** Blood biochemical values of SMAP8 mice fed with fermented Tilapia by-product and *Monostroma nitidum* oligosaccharide complex combined with exercise training.

Group	A	B	C	D	E
Glucose (mg/dL)	106.18 ± 8.40	122.32 ± 4.67	122.93 ± 2.66	121.76 ± 5.94	123.35 ± 5.60
Total protein (g/dL)	5.40 ± 0.15	5.42 ± 0.29	5.47 ± 0.29	5.55 ± 0.33	5.59 ± 0.29
Albumin (g/dL)	3.28 ± 0.20	3.37 ± 0.18	3.34 ± 0.22	3.53 ± 0.14	3.49 ± 0.19
Triglyceride (mg/dL)	98.60 ± 6.11	105.47 ± 5.09	112.13 ± 3.50	108.27 ± 6.59	110.80 ± 3.64
CHOL (mg/dL)	119.73 ± 4.28	120.20 ± 2.88	113.27 ± 2.79	122.73 ± 4.20	115.80 ± 3.01
ALP (U/L)	76.60 ± 4.30	79.00 ± 2.78	81.13 ± 1.52	76.87 ± 4.02	78.20 ± 2.87
GOT (U/L)	135.92 ± 4.90	133.54 ± 4.31	130.90 ± 2.19	124.85 ± 2.08	125.03 ± 4.89
GPT (U/L)	57.47 ± 3.13	56.80 ± 4.29	53.20 ± 1.71	56.67 ± 2.65	54.93 ± 3.04
Creatine kinase (U/L)	275.53 ± 7.30	287.40 ± 8.83	278.27 ± 5.29	260.60 ± 9.56	263.40 ± 6.41

Notes: Data are expressed as the mean ± S.E.M. (*n* = 15) and analyzed by one-way ANOVA. A: Gr-A, B: Gr-B, C: Gr-C, D: Gr-D, E: Gr-E.

**Table 3 nutrients-13-01688-t003:** Effect of the exercise test on lactate in SMAP8 mice fed with fermented Tilapia by-product and *Monostroma nitidum* oligosaccharide complex combined with ET.

Group	A	B	C	D	E
Lactate—Before swimming (mmol/L)	3.16 ± 0.60	3.21 ± 0.52	3.00 ± 0.43	3.17 ± 0.34	3.13 ± 0.39
Lactate—After swimming (mmol/L)	5.87 ± 0.56 ^a^	4.67 ± 0.49 ^b^	4.39 ± 0.46 ^b^	4.36 ± 0.19 ^b^	4.23 ± 0.30 ^b^
Lactate—After 20 min rest (mmol/L)	5.07 ± 0.55	3.87 ± 0.47	3.71 ± 0.44	3.51 ± 0.26	3.50 ± 0.35
Production rate of serum lactate	2.79 ± 0.90 ^a^	1.37 ± 0.61 ^ab^	1.03 ± 0.42 ^b^	0.58 ± 0.15 ^b^	0.61 ± 0.19 ^b^
Clearance rate of serum lactate	0.15 ± 0.03	0.18 ± 0.04	0.17 ± 0.02	0.20 ± 0.04	0.18 ± 0.04

Notes: Data are expressed as the mean ± S.E.M. (*n* = 15) and analyzed by one-way ANOVA. Means in the same row followed by different letters (a, b) are significantly different (*p* < 0.05). A: Gr-A, B: Gr-B, C: Gr-C, D: Gr-D, E: Gr-E.

## Data Availability

The data provided in this study are available on request from Corresponding Author. Due to patent pending, privacy and ethical restrictions, this data is not publicly available.

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
