# Peer review of "Anti-Fatigue Effect of a Dietary Supplement from the Fermented By-Products of Taiwan Tilapia Aquatic Waste and Monostroma nitidum Oligosaccharide Complex"

_nutrients, 2021, doi:10.3390/nu13051688_

Round 1
Reviewer 1 Report
This study investigated the effects of a supplement on exercise performance and anti-fatigue effect. The data is fine for presentation but clarification is needed throughout on a number of items in the experimental design. My biggest concern is the dose of the supplement.
1) How was the dose of the supplement determined? How much BCAA and other nutrients do they contain? Detailed information should be presented as Table.
2) Why didn't the authors include skeletal muscle weight and muscle glycogen data? These indicators are indispensable for this paper.
3) Most of discussions are based on previous researches of BCAA supplementation. However, readers of this paper can not follow the logic without the information about nutrient compositions of this supplement. Howe about the possible contribution of DHA?
4) Blood lactate is not an appropriate marker of fatigue.
https://pubmed.ncbi.nlm.nih.gov/15308499/
https://pubmed.ncbi.nlm.nih.gov/11579151/
https://pubmed.ncbi.nlm.nih.gov/15131240/
Author Response
[May 1st , 2021] Manuscript ID: nutrients-1177202
Dear reviewer 1:
I am very grateful to your comments for the manuscript. According with your advice, we amended the relevant part in manuscript. Some of your questions were answered below.
Question 1:
How was the dose of the supplement determined? How much BCAA and other nutrients do they contain? Detailed information should be presented as Table.
Author response:
The material of Taiwan Tilapia waste product used in this study was provided by National Taiwan Ocean University (NTOU) under the project funded by Council of Agriculture, Executive Yuan R.O.C. (Republic of China, Taiwan). The dose of supplement was converted by suggested dose from NTOU. Taiwan Tilapia extract is in powder form and needs to be diluted with ddH2O before use.
The recommended dose of Taiwanese Tilapia extract is 67 mg/kg body weight/day. The dose for mice used here was converted from a human equivalent dose (HED) based on body surface area by the following formula from the US Food and Drug Administration. We take 12.3 times the recommended intake for adults as 1 times the dose of mice.
1 times dose of mice : 67mg/kg BW/day x 12.3 = 824 mg/kg BW/day.
Another 0.5 times and 2 times dose groups:
0.5 times dose of mice : 67mg/kg BW/day x 12.3 x 0.5 = 412 mg/kg BW/day
2 times dose of mice : 67mg/kg BW/day x 12.3 x 2 = 1648 mg/kg BW/day
Question 2:
2) Why didn't the authors include skeletal muscle weight and muscle glycogen data? These indicators are indispensable for this paper.
Author response:
One of the main methods to remove excess lactic acid from laborious exercise is to convert lactic acid into glucose through gluconeogenesis, and the excess glucose will be retained as liver glycogen [Ref 1]. The breakdown of glycogen in muscles provides the initial source of energy required for exercise. After strenuous exercise, it may be exhausted, and in the later stages the energy will come from liver glycogen [Ref 2]. Therefore, in this experiment, we will preliminarily detect liver glycogen and exam with blood lactate, blood urea nitrogen and exercise performance capabilities, exhaustive swimming test, forelimb grip strength test.
Ref:
1.Qiang Liu and Yajun Li. Anti-fatigue Effects of Polysaccharide from Angelica Sinensis. IOP Conf. 2018 Ser.: Mater. Sci. Eng. 392 052011.
2.Anand T, Phani Kumar G, Pandareesh MD, Swamy MS, Khanum F and Bawa AS: Effect of bacoside extract from Bacopa monniera on physical fatigue induced by forced swimming. Phytother Res. 26:587–593. 2012.
Question 3:
3) Most of discussions are based on previous researches of BCAA supplementation. However, readers of this paper can not follow the logic without the information about nutrient compositions of this supplement. Howe about the possible contribution of DHA?
Author response:
A study about Santé Premium Silver Perch Essence (SPSPE) showing anti-fatigue effect with rich BCAA content (7.14mg/mL). SPSPE could efficiently delay swimming fatigue through sparing of liver glycogen and attenuation of plasma thiobarbituric acid reactive substances, myoglobin induction by exhaustive exercise. [Ref 1].
The results of this research project sponsored by the Council of Agriculture, Executive Yuan R.O.C. are in the process of applying for a patent, and some of the ingredients have not been fully disclosed. The total BCAA content of the functional component of the compound of fermented By-products of Taiwan Tilapia aquatic waste and Monostroma nitidum oligosaccharide complex is 218 mg/g.
Therefore, we have given supplements (complex of Taiwan Tilapia by-products fermented with Monostroma nitidum oligosaccharides) combined with exercise training, and we have also seen the effect of improving anti-fatigue.
Based on your review opinion, we have added information about the content of BCAA in the body of the manuscript (Line 80-82).
In past literature, it was found that DHA is helpful in alleviating exercise fatigue. More evidence shows that DHA and EPA have anti-fatigue effects. Double-blind studies have shown that in untrained individuals, fish oil supplementation can inhibit muscle damage and reduce delayed-onset muscle soreness (DOMS) caused by excessive exercise load. [Ref 2].
This test material uses Taiwan Tilapia fish skin, bones, and vegetable protein of Monostroma nitidum as raw materials and fermentation with lactic acid bacteria and its extract. The remaining skin, bone, tail, and head of the Tilapia fillets are reused and supplemented with Monostroma nitidum, in order to fully utilize waste product of fish fillets processing and increase the value of waste products.
This time there is not focused on the discussion of DHA, and we will continue to pay attention to this aspect in the future.
Ref :
- Chen CY, Yuen HM, Lin CC, Hsu CC, Bernard JR, Chen LN, Liao YH, Tsai SC. Anti-fatigue Effects of Santé Premium Silver Perch Essence on Exhaustive Swimming Exercise Performance in Rats. Front Physiol. 2021 Mar 22;12:651972.
2.Wang CC, Shi HH, Zhang LY, Ding L, Xue CH, Yanagita T, Zhang TT, Wang YM. The rapid effects of eicosapentaenoic acid (EPA) enriched phospholipids on alleviating exercise fatigue in mice. RSC Adv. 2019, 9, 33863
Question 4:
4) Blood lactate is not an appropriate marker of fatigue.
Author response:
We refer to some literature and found that serum lactate is feasible as an evaluation of anti-fatigue effect after exercise. Certain biochemical parameters have been published in the journal literature in the past to assess the degree of fatigue and post-exercise injury caused by exercise, such as lactate, BUN, CK, ammonia and free fatty acid [Ref 1, 2]. Blood lactate is derived from the anaerobic metabolism of glucose during exercise [Ref 3].
Ref:
- Chen HC, Huang CC, Lin TJ, Hsu MC, Hsu YJ. Ubiquinol Supplementation Alters Exercise Induced Fatigue by Increasing Lipid Utilization in Mice. Nutrients 2019 Oct 23;11(11):2550.
2.Hsu, Y.J.; Huang, W.C.; Lin, J.S.; Chen, Y.M.; Ho, S.T.; Huang, C.C.; Tung, Y.T. Kefir Supplementation Modifies Gut Microbiota Composition, Reduces Physical Fatigue, and Improves Exercise Performance in Mice. Nutrients 2018, 10, 862.
3.Kim, K.M.; Yu, K.W.; Kang, D.H.; Suh, H.J. Anti-stress and anti-fatigue effect of fermented rice bran. Phytother. Res. 2002, 16, 700–702.
Thank you for your questions and comments.
All corrections have been completed and we look forward to hearing any information about the submission
Sincerely,
Corresponding author
Liang-Hung Lin
Taichung Tzu Chi Hospital, Buddhist Tzu Chi Medical Foundation
Email address: linlianghung@gmail.com

Reviewer 2 Report
The article is well written, well designed and I have no remarks
Author Response
[May 1st , 2021] Manuscript ID: nutrients-1177202
Dear reviewer 2:
Thank you for taking the time to review our manuscript during daily busy work.
All corrections have been completed and we look forward to hearing any information about the submission
Sincerely,
Corresponding author
Liang-Hung Lin
Taichung Tzu Chi Hospital, Buddhist Tzu Chi Medical Foundation
Email address: linlianghung@gmail.com

Reviewer 3 Report
Here are my comments
Introduction:
1. The first sentence states that studies, but what you have cited is one study. Additionally, the meaning of fatigue has changed significantly since 2018. Please refer to the work by Loy, et al (2018) and subsequent publications by Boolani, et al on how fatigue has been re-defined.
2. Are there any other studies that have used the Tilapia BCAAs in fatigue? If not, you need to provide a theoretical justification as to why they would work. You need to provide some justification for why this product was used to see if there are anti-fatiguing effects during exercise.
Methodology:
Did you perform an a priori power analysis to determine group sizes?
Since the results are presented as Groups A-E can you please describe the interventions as Groups A-E or you should change the results.
In the methodology, it would also help if you identified each group so that when the reader scrolls down to the tables in the results it is easier for the reader to interpret.
For the statistical analysis section, why did you not use a repeated measures ANOVA?
For the lactate data that you have collected a repeated measures ANOVA would be the most appropriate test to perform. The repeated measures ANOVA will allow you to test changes in lactate over time between groups.
Discussion
I feel that there are statistical questions regarding this manuscript therefore I am unable to properly evaluate the discussion section, as the results may be different thus changing the discussion.
Author Response
[May 1st , 2021] Manuscript ID: nutrients-1177202
Dear reviewer 3:
I am very grateful to your comments for the manuscript. According with your advice, we amended the relevant part in manuscript. Some of your questions were answered below.
Question 1:
Introduction:
- The first sentence states that studies, but what you have cited is one study. Additionally, the meaning of fatigue has changed significantly since 2018. Please refer to the work by Loy, et al (2018) and subsequent publications by Boolani, et al on how fatigue has been re-defined.
- Are there any other studies that have used the Tilapia BCAAs in fatigue? If not, you need to provide a theoretical justification as to why they would work. You need to provide some justification for why this product was used to see if there are anti-fatiguing effects during exercise.
Author response:
The goal of our research is focused on the effect of nutritional supplementation and exercise training on the improvement of physical fatigue [Ref 1]. Physical fatigue is usually defined as a reversible reduction in performance during exercise. fatigue is any decline in muscle performance associated with muscle activity. This is particularly clear when maximum isometric force is measured in repeated tetani [Ref 2]. The known fatigue mechanism is related to the availability of metabolic fuel and waste accumulation [Ref 3].
Thanks to reviewer’s questions and comments, we have revised the text of the article and added cited journals.
Ref:
- Chen HC, Huang CC, Lin TJ, Hsu MC, Hsu YJ. Ubiquinol Supplementation Alters Exercise Induced Fatigue by Increasing Lipid Utilization in Mice. Nutrients. 2019 Oct 23;11(11):2550.
- Allen, D.G.; Lamb, G.D.; Westerblad, H. Skeletal muscle fatigue: Cellular mechanisms. Physiol. Rev. 2008, 88, 287–332.
- Kawamura, T.; Muraoka, I. Exercise-induced oxidative stress and the effects of antioxidant intake from a physiological viewpoint. Antioxidants 2018, 7, 119.
In recent years, there has been great interest in using bioactive peptides as health foods and functional supplement [Ref 1]. Until recently, bioactive peptides have shown a variety of therapeutic effects, such as protein hydrolysates from meat and fish, which have antihypertensive, antioxidant, antimicrobial and antiproliferative effects [Ref 2,3]. However, research on Tilapia bioactive peptides and anti-fatigue effect is rare.
Ref:
1.Ashraf SA, Adnan M, Patel M, Siddiqui AJ, Sachidanandan M, Snoussi M, Hadi S. Fish-based Bioactives as Potent Nutraceuticals: Exploring the Therapeutic Perspective of Sustainable Food from the Sea. Mar Drugs. 2020 May 18;18(5):265.
2.Ryan, J.T.; Ross, R.P.; Bolton, D.; Fitzgerald, G.F.; Stanton, C. Bioactive peptides from muscle sources: Meat and fish. Nutrients 2011, 3, 765–791.
3.Chakrabarti, S.; Guha, S.; Majumder, K. Food-Derived Bioactive Peptides in Human Health: Challenges and Opportunities. Nutrients 2018, 10, 1738.
Question 2:
Methodology:
Did you perform an a priori power analysis to determine group sizes? Since the results are presented as Groups A-E can you please describe the interventions as Groups A-E or you should change the results. In the methodology, it would also help if you identified each group so that when the reader scrolls down to the tables in the results it is easier for the reader to interpret.
Author response:
We haven’t conducted priori power analysis yet. The experimental animals used in this experiment are senescence-accelerated prone mouse (SAMP8), a special aging-promoting mouse. At present, there are few literatures on the use of SAMP8 for anti-fatigue tests. Therefore, using such group sizes, we would like to observe whether there is a consistency in performance between groups.
According to your suggestions, we have regrouped and marked all the groups to make it easier for readers to read and understand between the article and the table (Line 151-164, Table 1-2).
Question 3 & 4:
For the statistical analysis section, why did you not use a repeated measures ANOVA? For the lactate data that you have collected a repeated measures ANOVA would be the most appropriate test to perform. The repeated measures ANOVA will allow you to test changes in lactate over time between groups.
Discussion: I feel that there are statistical questions regarding this manuscript therefore I am unable to properly evaluate the discussion section, as the results may be different thus changing the discussion.
Author response:
In some literature, lactate can be used as a feasible method to evaluate the anti-fatigue effect after exercise. In the past, certain biochemical parameters have been published in the journal literature to assess the degree of fatigue and post-exercise injury caused by exercise, such as lactic acid, BUN, CK, ammonia and free fatty acids [References 1, 2]. Blood lactate level comes from the anaerobic metabolism of glucose during exercise [Reference 3]. We refer to the statistical methods of the following documents, and use a statistical method of one-way ANOVA to observe the changes in the lactate level between each group. The results showed that in all the study groups that received exercise training (Gr-B,C,D,E), the serum lactic acid level was lower, and the clearance rate of serum lactic acid was higher. Group without exercise training did not show such result. This indicated shows that in addition to giving the mixture of Tilapia by-product fermented with Monostroma nitidum oligosacchride combined with physical exercise can slow down the formation of lactate.
Reference:
- Chen HC, Huang CC, Lin TJ, Hsu MC, Hsu YJ. Ubiquinol Supplementation Alters Exercise Induced Fatigue by Increasing Lipid Utilization in Mice. Nutrients 2019 Oct 23;11(11):2550.
2.Hsu, Y.J.; Huang, W.C.; Lin, J.S.; Chen, Y.M.; Ho, S.T.; Huang, C.C.; Tung, Y.T. Kefir Supplementation Modifies Gut Microbiota Composition, Reduces Physical Fatigue, and Improves Exercise Performance in Mice. Nutrients 2018, 10, 862.
3.Kim, K.M.; Yu, K.W.; Kang, D.H.; Suh, H.J. Anti-stress and anti-fatigue effect of fermented rice bran. Phytother. Res.2002, 16, 700–702.
Thank you for your questions and comments.
All corrections have been completed and we look forward to hearing any information about the submission
Sincerely,
Corresponding author
Liang-Hung Lin
Taichung Tzu Chi Hospital, Buddhist Tzu Chi Medical Foundation
Email address: linlianghung@gmail.com

Round 2
Reviewer 3 Report
Thank you for addressing all of my comments. The statistical analysis needs to be explained in the manuscript to reflect that you compared the amount of change between groups rather than just differences between groups.
Additionally, can you please add a limitations section to this manuscript.
Author Response
[May 8th, 2021] Manuscript ID: nutrients-1177202
Dear reviewer3:
I am very grateful to your comments for the manuscript. According with your advice, we amended the relevant part in manuscript. Some content of the manuscript been partially edited according to your suggestions. We highlight the changes as red/yellow-colored marks.
Line 391-392
3.Results
3.4.3 Blood Lactate Level
We refer to the statistical methods of the following documents, and use a statistical method of one-way ANOVA to observe the changes in the lactate level between each group.
Line 397-399
After the exercise test, the lactate concentration were significantly lower in the Gr-B, Gr-C, Gr-D and Gr-E groups than in the Gr-A group, by 20.4%, 25.2%, 25.7%, and 27.9% (all p<0.05), respectively.
Line 404-409
The results showed that in all the study groups that received exercise training (Gr-B, Gr-C, Gr-D, Gr-E), the serum lactate level was lower, and the clearance rate of serum lactate was higher. Groups without exercise training did not show such result. This indicates that in addition to giving the mixture of Tilapia by-product fermented with Monostroma nitidum oligosacchride combined with physical exercise can slow down the formation of lactate.
Line 581-586
4.Discussion
Our results suggest that the combination of physical training and consumption of dietary supplementation mixture of fermented Tilapia by-products and Monostroma nitidum oligosaccharides is an effective method for anti-fatigue. We suggest that this could be a new sport nutrition supplement for reducing fatigue. Therefore, further research on anti-fatigue effect of some micronutrients in Tilapia aquatic waste, such as DHA, will further understand its physiological role and mechanism in our next study.
Thank you for your questions and comments.
All corrections have been completed and we look forward to hearing any information about the submission
Sincerely,
Corresponding author
Liang-Hung Lin
Taichung Tzu Chi Hospital, Buddhist Tzu Chi Medical Foundation
Email address: linlianghung@gmail.com
